# Thermochemical Characterization of Eight Seaweed Species and Evaluation of Their Potential Use as an Alternative for Biofuel Production and Source of Bioactive Compounds

**DOI:** 10.3390/ijms23042355

**Published:** 2022-02-21

**Authors:** Lucía Cassani, Catarina Lourenço-Lopes, Marta Barral-Martinez, Franklin Chamorro, Pascual Garcia-Perez, Jesus Simal-Gandara, Miguel A. Prieto

**Affiliations:** 1Nutrition and Bromatology Group, Department of Analytical and Food Chemistry, Faculty of Food Science and Technology, Universidade de Vigo, Ourense Campus, E32004 Ourense, Spain; luciavictoria.cassani@uvigo.es (L.C.); c.lopes@uvigo.es (C.L.-L.); marta.barral@uvigo.es (M.B.-M.); franklin.noel.chamorro@uvigo.es (F.C.); pasgarcia@uvigo.es (P.G.-P.); 2Instituto de Investigaciones en Ciencia y Tecnología de Materiales (INTEMA, CCT-CONICET), Av. Colón 10850, Mar del Plata 7600, Argentina; 3Department for Sustainable Food Process, Università Cattolica del Sacro Cuore, Via Emilia Parmense 84, 29122 Piacenza, Italy; 4Centro de Investigação de Montanha (CIMO), Instituto Politécnico de Bragança, Campus de Santa Apolonia, 5300-253 Bragança, Portugal

**Keywords:** macroalgae, nutritional composition, alternative bioenergy resource, functional ingredients, antioxidant activity, anti-inflammatory activity

## Abstract

Algae are underexplored resources in Western countries and novel approaches are needed to boost their industrial exploitation. In this work, eight edible seaweeds were subjected to their valorization in terms of nutritional characterization, thermochemical properties, and bioactive profile. Our results suggest that seaweeds present a rich nutritional profile, in which carbohydrates are present in high proportions, followed by a moderate protein composition and a valuable content of ω-3 polyunsaturated fatty acids. The thermochemical characterization of seaweeds showed that some macroalgae present a low ash content and high volatile matter and carbon fixation rates, being promising sources for alternative biofuel production. The bioactive profile of seaweeds was obtained from their phenolic and carotenoid content, together with the evaluation of their associated bioactivities. Among all the species analyzed, *Porphyra purpurea* presented a balanced composition in terms of carbohydrates and proteins and the best thermochemical profile. This species also showed moderate anti-inflammatory activity. Additionally, *Himanthalia elongata* extracts showed the highest contents of total phenolics and a moderate carotenoid content, which led to the highest rates of antioxidant activity. Overall, these results suggest that seaweeds can be used as food or functional ingredient to increase the nutritional quality of food formulations.

## 1. Introduction

Algae have traditionally been used as sea vegetables, animal feed, fertilizers, and exploited by the phycocolloids industry for the extraction of agar-agar, carrageenan, and alginate [1]. In spite of being a billionaire industry with a market size valued at USD 6 billion per year, with China, Japan, and South Korea being the main consumers [2], the worldwide algae production is still restricted to the cultivation of edible species and for the phycocolloids industry [3]. The most widely consumed species belong to the *Undaria* (wakame) (Ochrophyta, Phaeophyceae), *Porphyra* (nori) (Rhodophyta), and *Laminaria* (kombu) (Ochrophyta, Phaeophyceae) genera [4]. Within the green seaweed phylum, *Ulva rigida* (sea lettuce) is also used for human consumption [5]. Other works have highlighted the valuable nutritional composition of different seaweed classes (green, brown, and red) as fresh or dried vegetables, even suggesting their incorporation as ingredients in several processed foods. From a nutritional point of view, seaweeds are attractive for their high protein content (up to 47% of dry weight in red species), being comparable to meat and dry legumes. Their low-fat content is also remarkable with a healthy ratio of fatty acids ω6/ω3, and high dietary fiber content, which can reach 50% of dry weight in brown species with soluble fiber as the predominant fraction [6]. From a technological perspective, seaweeds present many functional properties, such as water retention capacity, oil holding capacity, swelling, etc, associated with their protein and dietary fiber content that supported their incorporation into farinaceous products (bread, noodles, pasta, cake, biscuit, cookies, etc. [6]).

In addition, seaweeds are an important and underexploited source of bioactive compounds, including sulfated polysaccharides (fucoidan), polyphenols (phlorotannins), and natural pigments (fucoxanthin, β-carotene, chlorophylls, xanthophylls), among others [7,8,9]. These compounds have proved to have biological activities, acting as antioxidant, antimicrobial, anti-inflammatory, anti-tumoral, antifungal, antiviral, and neuroprotective agents, and thus, can be used as functional ingredients in many innovative and technological applications in food, cosmeceutical, and pharmaceutical industries [8,9].

On the other hand, seaweeds have been gaining attention as an alternative renewable source for “third-generation” biofuels production. Seaweed biomass is an attractive feedstock since it has many advantages over terrestrial plant biomass including higher growth rate, higher rate of carbon dioxide fixation by photosynthesis, absence of hemicellulose and lignin, which facilitates the thermal depolymerization, no requirements of arable land for cultivation, and the ability to grow in saltwater or wastewater [10]. Therefore, this offers new opportunities to exploit these promising natural resources for biofuel production either from wastes generated after phycocolloids and food production or raw biomass [11].

Anaerobic digestion, fermentation, transesterification, liquefaction, and thermochemical conversion can convert seaweed biomass into valuable energy products, for instance, biogas, bioethanol, bio-methane, biodiesel, and bio-oils [12]. Among different pathways, thermochemical conversion is one of the most extensively studied processes in recent years since different biofuel products in solid, liquid, and gaseous states can be generated from seaweed biomass [13]. Pyrolysis is a thermochemical process that involves the thermal degradation of the biomass components in an inert atmosphere and generates bio-oil, biochar, and gas useful for energy production [13]. Other works have investigated the thermochemical conversion of microalgae biomass to biofuel however, less attention has been paid to the use of macroalgae for that purpose.

In this context, four brown edible seaweeds (*Himanthalia elongata* (L.) S.F.Gray, *Laminaria ochroleuca* de la Pylaie, *Undaria pinnatifida* (Harvey) Suringar, and *Saccharina latissima* (Linnaeus) C.E. Lane, C. Mayes, Druehl and G.W. Saunders [formerly *Laminaria saccharina* (L.) Lamouroux], two green (*Ulva rigida* C. Agardh and *Codium tomentosum* Stackhouse) and two red species (*Porphyra purpurea* (Roth) C. Agardh and *Palmaria palmata* (L.) F. Weber and D. Mohr) (Figure 1) were selected to be valorized in terms of nutritional characterization, thermochemical properties, and bioactive profile, all of them widely distributed throughout Atlantic coastlines and still largely unexplored. To this aim, such seaweeds were evaluated as (1) an alternative renewable resource for biofuel production based on their thermochemical conversion through the pyrolysis pathway and, (2) as a source of bioactive compounds with antioxidant and anti-inflammatory activities.

## 2. Results and Discussion

### 2.1. Nutritional Composition of Seaweeds

The chemical composition of seaweeds is shown in Table 1. The protein content of the studied macroalgae ranged from 9.81 to 34.79 g/100 g DW and red algae obtained the highest values. These results are in line with Sanchez-Machado [14] who also reported that the highest protein content corresponded to *Porphyra purpurea* and Taboada [15] who found 33.2 g protein/100 g DW for that species [15]. Indeed, the protein values obtained for red macroalgae are comparable to other traditional protein source foods, such as soybean, cereals, eggs, and fish [16]. In contrast, brown algae showed the lowest protein content except for *Undaria pinnatifida* whose values were like green macroalgae. This was an expected result since Garcia-Vaquero [16] concluded that brown seaweeds had low protein content and within this group, *U. pinnatifida* sample was the species that showed the highest value (up to 24 g/100 g DW). Our results are also comparable to those by Taboada [15] who reported 16.8 g protein/100 g DW for *U. pinnatifida*. All the analyzed seaweeds are a reliable source of carbohydrates, which ranged from 41.54 to 65.92 g/100 g DW, and brown algae showed the highest values. As polysaccharides synthesis could influence protein formation [16], the highest carbohydrate content observed in brown algae are in consonance with their previously reported low protein values. Likewise, Taboada [15] found that the carbohydrate content of *U. pinnatifida* and *P. purpurea* was 53.9 and 44.5 g/100 g DW, respectively. Seaweed carbohydrates are mainly composed of polysaccharides (in the sulfated and non-sulfated forms depending on the algae species) with a low proportion of di- and monosaccharides [17]. The main polysaccharides from brown algae are alginate, fucoidans, and laminarin [18]. Red seaweeds are rich in agar, carrageenan, sulfated galactans, xylans, and xylogalactans, while green algae contain mainly ulvan [18]. Most of these compounds have been widely explored for their hydrocolloid properties amplifying their use as emulsifiers, stabilizers, and viscosity-controlling ingredients in several applications [19]. In addition, other works have ascribed important bioactivities to macroalgae polysaccharides, such as antioxidant, anti-inflammatory, antitumor, antidiabetic, and antimicrobial activities, suggesting that seaweeds open the door to innovative and technological applications in food, nutrition, and pharmaceutical industries [20].

The total lipid content was very low in most of the studied species and ranged from 0.33 to 4.22 g/100 g DM. Sanchez-Machado [14] reported that the total lipid content of seaweeds is low and represents less than 4% of dry weight. Among the studied seaweeds, the highest lipid content was observed in *Palmaria palmata* (4.22%), and this was comparable with results found by Lopes [21]. Regarding brown algae species, similar lipid contents in *Himanthalia elongata*, *U. pinnatifida* and *Laminaria ochroleuca* (1.74, 1.24, and 0.72 g/100 g dry weight, respectively) were observed with respect to the values reported by Sanchez-Machado [14].

The fatty acids (FAs) composition of the studied seaweeds is displayed in Table 2. Saturated FAs had the main contribution to the total FAs content in all the analyzed species (43.95–64.12%) except for *Ulva rigida* (31.84%) and *U. pinnatifida* (31.70%). Palmitic acid (C16:0) was the most abundant compound within this group and *P. palmata* was the species that showed the highest relative content (42.05%). In addition, red algae showed the highest stearic acid content (C18:0). Similar results were found by Sanchez-Machado [14] who reported that saturated FAs were predominant compounds in red macroalgae. Such a result agrees with our results, where the red seaweed *P. palmata* exhibited the highest content of saturated FAs (Table 2). Regarding monounsaturated FAs (MUFAs), oleic acid (C18:1) was the most abundant MUFA in all the analyzed species and UR showed the main contribution to the total MUFAs obtaining the highest value (29.51% of total FAs). In line with these results, Yaich [22] found that palmitic and oleic acid in *U. rigida* represented about 76% of the total FAs. Concerning polyunsaturated FAs (PUFAs), *U. pinnatifida* showed a predominant contribution to the total PUFAs (44.92%) while for the rest of the studied species, PUFAs content ranged from 13.99 to 30.00% (Table 2). The same tendency was reported by Sanchez-Machado [14] who also found that *U. pinnatifida* showed a significant contribution to the total PUFAs. Linoleic acid (C18:2 ω6), an essential fatty acid, was present in all the analyzed seaweeds and brown species (except *Saccharina latissima*) were those that exhibited the major content. Another essential fatty acid of great interest is linolenic acid (C18:3 ω3) which was also present in all the studied macroalgae, and *U. pinnatifida* showed the highest value (24.72% of total FAs) followed by *U. rigida* (14.25%) an *H. elongata* (11.46%). These results show that *U. pinnatifida* is a promising source rich in ω3 fatty acids. On the other hand, docosahexaenoic acid (DHA, C22:6 ω3), especially interesting for its contribution to infant brain development, was only detected in *P. palmata* but in a low relative content [23,24].

Furthermore, interesting results were found for the balance between ω6 and ω3 FAs since all the analyzed seaweeds showed low ω6/ω3 ratio values, below the recommended range (2–5:1). Indeed, *P. palmata*, *Codium tomentosum*, *U. rigida*, and *U. pinnatifida* exhibited a <1 ratio, due to the higher ω3 FAs proportion. In the Western diet, the ω6/ω3 ratio ranges from 15:1 to 20:1 due to the high consumption of ω6 FAs from terrestrial plants’ oils and the insufficient intake of ω3 FAs [24]. In this regard, the inclusion of the studied seaweeds in the diet could potentially contribute to reducing the unbalanced PUFAs ratio and thus, improving the nutritional quality of the Western diet.

The micro- and macro-minerals composition of the studied seaweeds is presented in Table 3. Results suggested that all the analyzed seaweeds are a rich source of nutritionally relevant minerals such as iron, manganese, iodine, calcium, potassium, magnesium, and phosphorus. *U. rigida* sample recorded the highest values for Fe and Mn and were greater than those reported by Mæhre [24]. In addition, *U. rigida* was the highest source of Mg, which was also observed by Mæhre [24]. Higher content of macrominerals was observed for all the studied macroalgae when compared to values found for conventional vegetables. In addition, iodine, an essential micronutrient with a key contribution to the synthesis of thyroid hormones, was also present in all the analyzed seaweeds. In fact, great variability in iodine content between and within macroalgae classes was observed which ranged from 82 mg/kg (*P. purpurea*) to 5829 mg/kg (*L. ochroleuca*). These results were in line with those compiled by Holdt [25] who also stated that *Laminaria* sp. can accumulate iodine up to 30,000 times the concentration found in seawater. In this context, an excessive consumption of *L. ochroleuca* and *S. latissima* could lead to undesirable effects.

In conclusion, variations in nutritional composition between algae species were seen. These differences can be attributed to phyla, harvesting season, environmental conditions, and geographical location.

### 2.2. Performance of Seaweeds as an Alternative Bioenergy Resource

Table 4 depicts the proximate and ultimate analyses of the studied seaweeds. Concerning the proximate analysis, differences in moisture content between algae species were seen and ranged from 5.9% for *C. tomentosum* to 14.9% for *S. latissima*. Although these results are comparable to values reported for other terrestrial biomasses, a low moisture content of the feedstock is preferred for the pyrolysis process [26].

Ash content differed between seaweeds species (15.1–38.9%) and was higher than other biomass generally used for pyrolysis processes (e.g., 3.8–16% for leaves of various trees, 6.7% for sugar cane bagasse, 1% for wood, and 0.8% for briquette) [26]. This result is in line with that previously mentioned about the higher mineral content of seaweed in comparison to terrestrial plants. The salt content of sea water and rocks, where macroalgae can be generally found, have a significant contribution to this effect [28]. In addition, the analyzed macroalgae species exhibited lower volatile matter content (41.3–61.8%) than the values reported for terrestrial energy crops (66.8–85.3%) [26]. This is attributed to the lack of lignocellulosic compounds (cellulose, hemicellulose, and lignin) in seaweeds composition which are structurally predominant in many terrestrial plants. By contrast, unique polysaccharides (previously mentioned) constitute the main carbohydrate fraction of seaweeds and thus, can be depolymerized easier than lignocellulosic plants [10,29]. For acting as a promising energy resource, biofuels should meet two important requirements: present a maximum of 20% of ash content to avoid operational problems associated with ash composition [26] (e.g., slagging, fouling, sintering, and corrosion) and exhibit a high volatile matter content to be more available to thermal degradation during pyrolysis process [26]. In this context, among the studied seaweeds, *P. purpurea* showed the lowest ash content and the highest volatile matter, suggesting that could be a potential alternative for biofuel production based on the pyrolysis pathway. In addition, a suitable treatment for removing the seaweeds’ ash content prior to conducting a pyrolysis process should be designed with a large-scale biofuel production perspective. The proportion of fixed carbon in the studied seaweeds (13.3–19.1%) are in line with values reported for other terrestrial biomasses, except for *U. rigida*, which showed the lowest fixed carbon value, <6% (Table 4). For biochar production, fixed carbon is needed to be used as carbonaceous materials during biomass pyrolysis [28]. Thus, based on this parameter, *P. palmata* could produce more biochar and other pyrolysis products than *U. rigida*.

Regarding ultimate analysis, seaweeds showed lower carbon and hydrogen content and higher nitrogen and oxygen proportion than values reported for other terrestrial biomasses, except for *P. purpurea* which showed carbon and hydrogen values comparable to content reported for sugar cane straw. For example, the basic elemental composition for typical biomasses used for combustion, such as sugar cane bagasse, wood, and briquette, ranges between 46.7–57.2% for C, 6.1–6.4% for H, 0–1.2% for N, and 41.5–45.5% for O [30]. Nitrogen is present in many biological compounds of seaweeds, such as proteins, chlorophyll, amino acids, and vitamins [30]. Higher nitrogen content could be problematic during the thermochemical conversion of seaweed biomasses since toxic and corrosive nitric oxides (NOx) could be released leading to a negative environmental impact [28].

The high heating values of the analyzed macroalgae were in the range 14.07–19.08 MJ/kg and were similar to values reported for terrestrial energy crops (17–20 MJ/kg) [29].

Figure 2 displays thermograms from TG, DTG, and DSC analyses that allowed finding the thermochemical behavior of seaweeds during the pyrolysis process. According to the TG curves, all the studied seaweeds showed similar thermochemical behavior, and three stages of biomass pyrolysis were identified. The first stage corresponds to dehydration and evaporation of highly volatile matter and occurred within the range of 20–150 °C. In this stage, small weight loss (in general 10%) for all samples was recorded. DTG curves confirmed that the biomasses dehydration occurred at 75–100 °C. Complementarily, from DSC analysis, an endothermic point of inflection at 75–100 °C for all samples was seen, which suggests that seaweeds absorbed heat to evaporate their moisture content (Figure 2). The second stage refers to devolatilization reactions and took place in the range of 175–600 °C. In this stage, the major biomass weight loss as volatile matter was recorded. According to the DTG curves, the maximum weight loss occurred at ca. 250 °C for all samples and was comparable to values reported by Kebelmann [30] for several macroalgal species. In general, the pyrolytic decomposition of the analyzed seaweeds took place at lower temperatures with respect to the values reported for lignocellulosic biomasses [29] (straws, grasses, woody biomass, etc.) since hemicellulose degrades within the range of 220–260 °C while cellulose degrades at 315–390 °C [30]. Thus, the significant weight loss observed for the analyzed seaweeds is mainly attributed to the marine polysaccharides’ decomposition, as previously observed by other authors [31]. In addition, a less intense point of inflection at 320 °C was detected in the DTG curves of some samples, which could be ascribed to the protein content degradation [29]. As was expected, the organic material degradation follows an exothermic behavior, and thus two peaks at 175 °C and another at 250–300 °C was seen in DSC curves for all samples, which was more intense for *P. purpurea* (Figure 2G), *U. rigida* (Figure 2A), and *U. pinnatifida* (Figure 2C). Differences seen in decomposition rates obtained for all seaweeds can be attributed to differences in their chemical composition and natural structure.

The third stage corresponds to decomposition of volatile matter released in the previous stage with remaining protein and carbonaceous solids [31]. According to the DTG curves, samples degradation occurred slowly and thus, the weight loss was low (Figure 2). This fact could be ascribed to the slow degradation of the formed residue (biochar) [28]. In addition, some samples showed another point of inflection at 700–800 °C in the DTG curves which was more intense for *U. rigida* (Figure 2A). This additional mass degradation could be attributed to the devolatilization of inorganic compounds [28], which probably determine the amount of biochar produced [31]. Consistently, an exothermic peak at 700 °C was seen in DSC curves for some macroalgae, and *U. rigida* recorded the maximum intensity.

### 2.3. Performance of Seaweeds as a Promising Source of Bioactive Compounds

Table 5 shows the extraction yield, total phenolic content, total carotenoid content, and total antioxidant activity obtained for aqueous–organic extracts of the studied seaweeds. Significant differences (*p* < 0.05) in the extraction yield were found between seaweed species (regardless of the seaweed class) ranging from 20.55 to 47.29% (Table 5, Appendix A). This can be ascribed to different polarities and solubility of the bioactive compounds in the mixture of aqueous methanolic and acetonic extracts, as well as to variations in chemical composition among species [32]. *S. latissima*, *L. ochroleuca*, and *C. tomentosum* showed the highest extraction yields (44.30–47.29%), suggesting a higher release of polar soluble compounds such as phytochemical compounds, polysaccharides, proteins, peptides, and organic acids from these species (Table 5, Appendix A) [32]. By contrast, *P. purpurea* exhibited the lowest extraction yield (~20%) and could indicate that bioactive compounds may have higher polarity [32], thus requiring more polar solvents for their extraction. However, the extraction yields obtained for all the analyzed seaweeds were higher than the values reported in the literature [33]. Many works have proposed that the extraction procedure of bioactive compounds from biological tissues should combine at least two extraction steps and use aqueous organic solvents with different polarities to extract bioactive compounds with diverse chemical structures [34]. In this regard, the application of heat-assisted extraction in combination with aqueous methanolic and acetonic extracts resulted in an effective procedure for the recovery of bioactive compounds of different structures from seaweeds.

Regarding phenolic compounds, TPC for all the analyzed seaweed ranged from 366.48 to 3448.55 µg/g DM, and significant differences (*p* < 0.05) between species (regardless of the seaweed class) were observed (Table 5, Appendix A). While *P. purpurea* and *H. elongata* showed the highest TPC (~3 mg GAE/g DM), *U. rigida*, *U. pinnatifida*, and *L. ochroleuca* registered the lowest values with no statistical differences between them (<0.6 mg GAE/g DM). As it can be seen in some cases, the extraction yield should not be strictly related to the phenolics content released from seaweeds, thus suggesting the presence of other chemical constituents in the obtained extracts. In this regard, other constituents, such as proteins or reducing sugars in seaweed extracts, can also reduce the Folin–Ciocalteu reagent, overestimating the phenolic content [35]. In addition, other authors reported that the extraction yield may not be correlated with the phytochemical content of extracts [36]. Comparing our results with those values found by other authors is a challenging task due to different extraction procedures used (solvent, temperature, etc.) and the method of expressing results (mg GAE per g extract instead of g DM). Despite this, some similarities in TPC with other works were found [37]. Polyphenolic compounds such as phlorotannins, bromophenols, flavonoids, phenolic terpenoids, and mycosporine-like amino acids have been found in seaweeds [9]. While phlorotannins are the main polyphenolic group present in brown algae, bromophenols, flavonoids, phenolic terpenoids, and mycosporine-like amino acids were observed in red and green seaweeds [9].

Concerning total carotenoids, significant differences in TCC between seaweed species (regardless of the seaweed class) were also reported, ranging from 61.27 to 295.24 µg/g DM (Table 5, Appendix A). *U. rigida*, *C. tomentosum*, *L. ochroleuca*, and *P. purpurea* exhibited the highest values (>200 µg/g DM), while *S. latissima* and *P. palmata* presented the lowest TCC (<80 µg/g DM). These results were higher than those reported by other authors and differences can be attributed to the method used for quantification, the inherent characteristics of species, environmental conditions that algae are exposed to, geographical location, harvesting period, etc. [38].

Carotenoids are considered as the major seaweed pigments, including xanthophylls and carotenes. The main pigment found in brown algae is fucoxanthin which has been widely studied for its promising biological activities, acting as a potent antioxidant, cytoprotective, anticancer, anti-inflammatory, neuroprotective, antidiabetic agent [39]. In addition, β-carotene was also recorded in brown algae. Among red seaweeds, β-carotene, α-carotene, zeaxanthin, and lutein have been the main carotenoids reported [40]. For green macroalgae, the carotenoid composition includes β-carotene, lutein, violaxanthin, antheraxanthin, zeaxanthin, and neoxanthin [40]. In this regard, the analyzed seaweeds represent a valuable source of carotenoids with diverse chemical structures which make them interesting alternatives to the artificial colorants commonly used in the food industry, whose controversial safety issues cause rejection among health-concerned consumers. Besides their intense color, they have important health-related properties, being potentially used as functional ingredients.

Regarding DPPH scavenging activity, only *H. elongata* showed DPPH radical quenching ability among all the studied seaweeds, exhibiting *IC*_50_ values of 5.78 ± 0.26 mg/mL (Table 5). The model parameters and determination coefficient are shown in Appendix A. The rest of the samples were not able to achieve the 50% of radicals scavenging at the studied concentrations. This result suggests that other compounds (namely polysaccharides, proteins) also extracted from seaweeds were partially dissolved in the DPPH methanolic solution and then, interfered in the measurement method. However, better results were obtained for ABTS radical scavenging activity, since most of the analyzed seaweeds showed high radical quenching ability with effective concentrations ranging from 0.50 to 7.73 mg/mL (Table 5). The model parameters and determination coefficients are shown in Appendix A. For instance, *L. ochroleuca* (*IC*_50_, 0.50 mg/mL) and *H. elongata* (*IC*_50_, 0.70 mg/mL) showed the highest ABTS radical scavenging activity while green algae recorded the lowest values. Indeed, these values were comparable with effective concentrations found by Chakraborty [41] for α-tocopherol (*IC*_50_, 0.73 mg/mL), indicating that *L. ochroleuca* and *H. elongata* may have a similar ability to scavenge ABTS radical as compared to the commonly used antioxidant in the food industry. Based on these results, the high bioactive compounds content (TPC and TCC) of *H. elongata* could be related to its ABTS radical scavenging activities, suggesting that phenolic and carotenoid compounds may manage the radical deactivation.

Regarding the β-carotene bleaching inhibition assay, *P. purpurea* recorded the highest antioxidant activity among seaweeds as it was able to delay β-carotene oxidation by 50% up to 2401 min per mg of extract (Table 5). Similarly, *H. elongata*, *U. pinnatifida* and *U. rigida* also showed a reliable performance, causing an oxidation delay for 1679, 917, and 762 min per mg of extract, respectively. These results suggest that antioxidant compounds with lipophilic nature present in these samples could protect β-carotene when exposed to the linoleate free radicals and other free radicals produced in the system, therefore delaying lipid oxidation. In this context, these extracts could be further studied for their potential use as natural antioxidants in a real lipid-based system, such as food emulsions, as recently proposed by García-Pérez [42].

Concerning crocin bleaching inhibition assay, *P. purpurea* was again the species that recorded the highest antioxidant activity, delaying the crocin oxidation by 50% up to 211 min per mg of extract (Table 5). *H. elongata* was also able to prevent the crocin discoloration, extending its protection by 50% up to 145 min per mg of extract. The rest of the species showed low crocin protection since higher amounts of extract should be added to delay the crocin oxidation for a considerable period. This could indicate that antioxidant compounds from these species have lipophilic characteristics since they showed low responses when exposed to aqueous systems [43].

As can be seen, different behaviors of seaweed extract were observed between the four antioxidant activity assays studied. These variations can be attributed that the mixture of phytochemical structures present in seaweed extracts showed different solubility in each solvent used in the antioxidant activity assays (DPPH methanolic solution, ABTS ethanolic solution, β-carotene emulsion, and aqueous crocin solution). This indicates the complex phytochemical structures in seaweed extracts may employ different mechanisms to exert antioxidant activity. For instance, while *U. rigida* showed no DPPH and ABTS antiradical activity, this species was able to delay the β-carotene oxidation to a greater extent than the rest of the algae.

Concerning the anti-inflammatory and cytotoxic activity of seaweed extracts, results are displayed in Table 6. In the case of anti-inflammatory activity, only 4 seaweed species showed a moderate performance, being those from *P. purpurea* and *C. tomentosum* the most active extracts, exhibiting *IC*_50_ values of 193 and 264 mg/mL, respectively. However, these results are >96% lower than those of dexamethasone, an anti-inflammatory drug used as a positive control (Table 6). The highest activity found for *P. purpurea* could be motivated by the high rates of phenolic compounds and carotenoids previously described, as it has been proposed by other seaweed species and terrestrial plants [44,45]. Concerning the anti-inflammatory and cytotoxic activity of seaweed extracts, results are displayed in Table 6. In the case of anti-inflammatory activity, only four seaweed species showed a moderate performance, being those from *P. purpurea* and *C. tomentosum* the most active extracts, exhibiting *IC*_50_ values of 193 and 264 mg/mL, respectively. However, these results are >96% lower than those of dexamethasone, an anti-inflammatory drug used as a positive control (Table 6).

The highest activity found for *P. purpurea* could be motivated by the high rates of phenolic compounds and carotenoids previously described, as it has been proposed by other seaweed species and terrestrial plants [44,45]. In this regard, Lee [46] recently determined the anti-inflammatory mechanisms associated with *Porphyra* extracts, showing a multifaceted mode of action involving nitric oxide scavenging, the inhibition of pro-inflammatory mediators, and the induction of antioxidant enzymes. On the other hand, the results for the cytotoxic activity of seaweed extracts showed a negligible effect, with effective concentrations > 400 mg/mL (Table 6). This *a priori* negative result can be useful in the case of the Vero cell line, as it suggests that macroalgae do not show a cytotoxic effect towards healthy cell lines, which makes their incorporation in food, cosmetic and pharmaceutical products easier from a safety point of view. It is important to note that, despite the lack of activity in these extracts, there is wide evidence about the cytotoxic activity of pure compounds isolated from macroalgae, especially fucoidan, bromophenols, and fucoxanthin [47]. This opens a new perspective for the search of bioactive compounds from seaweeds, tackling the application of purification strategies with the aim of obtaining more active extracts.

## 3. Materials and Methods

### 3.1. Sample Collection

Eight common seaweed species from the three main phyla, four brown (Phaeophyceae), two red (Rhodophyta), and two green (Chlorophyta), were kindly provided by Algas Atlanticas AlgaMar S.L. (Pontevedra, Spain, http://www.algamar.com accessed 30 December 2021). Samples of macroalgae were received dry and without natural residues. Once at the laboratory, seaweeds were crushed until obtaining a homogeneous powder with a particle size lower than 2.5 mm. The resulting powders were stored in air-tight plastic bags protected from light at room temperature (25 °C) for further analysis. Figure 1 shows the seaweed species used in this study with their Spanish common names.

### 3.2. Nutritional Composition

#### 3.2.1. Lipid Content

Samples (2 g) were extracted with 100 mL of n-hexane at 68 °C for 3 h using an automatic Soxhlet Büchi Extraction SystemB-811 (Büchi Labortechnik AG, Flawil, Switzerland). Once the solvent was completely recovered using a rotary evaporator, the lipid content was gravimetrically determined. Results were expressed as g/100 g dry matter (DM).

#### 3.2.2. Fatty Acid Composition

The fatty acid (FA) composition was evaluated using gas chromatography with flame ionization detector (GC-FID). To carry this out, the lipid-rich extracts obtained by the Soxhlet unit (Section 2.2) were subjected to a derivatization procedure to obtain volatile fatty acid methyl esters (FAMEs). Briefly, 100 μL of lipid extracts were mixed with 2 mL of heptadecanoic acid (HAD) and 4 mL of n-hexane. The mixture was stirred for 1 min and then 0.5 mL of sulfuric acid (2% in methanol) was added. The resulting mixture was stirred for 5 more minutes and then, centrifuged at 3000 rpm for 5 min. Next, 2 mL of supernatant holding FAMEs were mixed with 5 mL of n-hexane and stored until analysis.

FAMEs-containing samples were analyzed using a gas chromatograph (Chromatographic System Agilent 7820A) equipped with a column Agilent HP-88 (60 m × 250 µm × 0.25 µm) and a flame ionization detector. The chromatographic run was performed using two temperature programs. The first one consisted of a ramp of 5 °C/min until reaching 220 °C and 15 min of holding time, while the second one used a temperature ramp of 40 °C/min until reaching 250 °C with a holding time of 2 min. Helium was used as carrier gas and was injected at 1 mL/min. Finally, FA identification was conducted using external standards, and their calibration curve was used to determine FA composition. Results were reported as % of total fatty acid content.

#### 3.2.3. Protein Content

The protein content was found according to the Dumas method. An Elemental Analysis Unit (FISONS Carlo Erba EA1108) with a CHNS Microanalyzer was used for the quantification of N_2_. Briefly, samples (5 mg) were subjected to flash combustion which led to complete and instantaneous oxidation of organic elements. The combustion gases (O_2_, CO_2_, H_2_O, N_2,_ and nitrogen oxides) were collected and passed through traps to remove O_2_, CO_2,_ and H_2_O. Then, nitrogen oxides were carried by helium over a copper catalyst to convert them into nitrogen. Next, thermal conductivity was found in the mixture, which emits an electrical signal proportional to nitrogen content. The detection limit was 10 ppm. A factor of 6.25 was used to convert mg N_2_ to protein and thus, results were expressed as g/100 g DM.

#### 3.2.4. Carbohydrate Content

Carbohydrate content (including total fiber) was determined by difference using the data obtained for lipid, protein, and ash content (Section 2.3).

#### 3.2.5. Micro- and Micromineral Composition

The concentration of microelements: iron (Fe), manganese (Mn), chromium (Cr), molybdenum (Mo), coper (Cu), zinc (Zn), and selenium (Se); and macroelements: calcium (Ca), potassium (K), magnesium (Mg), and phosphorus (P) were simultaneously analyzed by inductively coupled plasma optical emission spectrometry (ICP-OES) using a Perkin–Elmer Optima 4300 DV spectrometer (Shelton, CT, USA), equipped with an AS-90 autosampler, axial system, a high dynamic range detector and a cross-flow type nebulizer for pneumatic nebulization.

The ICP-OES assessment was performed following the procedure described by Millos [48]. Prior to ICP-OES analyses, samples (0.25 g) were digested with concentrated nitric acid (HNO_3_) and hydrogen peroxide using a Multiwave 3000 oven (Anton Paar, Graz, Austria), equipped with eight digestion vessels, according to the methodology proposed by Millos [48]. Next, elements were analyzed by ICP-OES (axial configuration) and the equipment operating conditions for ICP-OES used in this study were those reported by Millos [48]. For the quantification of micro- and macroelements, standard stock solutions with the addition of ^115^In as an internal standard were used to construct the corresponding calibration curves. Results were reported as mg/kg DM.

In turn, the concentration of iodine (I) was conducted by inductively coupled plasma-mass spectrometry (ICP-MS) using a Thermo Elemental X7 Series ICP-MS equipped with an ASX-520 autosampler (Omaha, NE, USA) and PlasmaLab Software. Samples (0.25 g) were digested with tetramethylammonium hydroxide (TMAH) in closed vessels. Both the operating conditions and ICP-MS measurement were carried out following the method proposed by Millos [49].

These studies were conducted in the Food Security and Sustainable Development Laboratory, Scientific and Technological Support Centre for Research (SSADS-CACTI, University of Vigo, Vigo, Spain).

### 3.3. Performance of Seaweeds as an Alternative Bioenergy Resource

The potential of seaweeds biomass as a resource for sustainable biofuel production was studied. For the conversion of biomass to fuel, a pyrolysis process was checked that is, dry seaweeds were heated in the absence of air and the thermal decomposition of the organic components was evaluated. Proximate and ultimate analyses were conducted to evaluate the chemical properties of seaweed biomasses in the pyrolysis process which allow inferring about their suitability for biofuel production.

#### 3.3.1. Proximate Analysis

Moisture content: The moisture content of samples was thermogravimetrically determined according to the UNE-EN 14774-1 standardized procedure, using a SETSYS Evolution apparatus (Setaram Instrumentation, Caluire-et-Cuire, France). Samples (7–15 mg) were heated to 105 °C at 10 °C/min under an inert nitrogen flow (30 mL/min) until constant weight.

Volatile matter: The volatile matter was thermogravimetrically determined according to the UNE-EN 15,148 standardized procedure in a SETSYS Evolution apparatus (Setaram Instrumentation, Caluire-et-Cuire, France). Water-free samples (7–15 mg) were heated between 105 and 600 °C at 10 °C/min under an inert nitrogen flow (30 mL/min) until constant weight.

Ash content: The ash content was also thermogravimetrically measured following the procedure described in UNE-EN 14775. Water- and volatile matter-free samples (7–15 mg) were combusted at 900 °C under oxidizing air atmosphere (30 mL/min) until constant weight.

Fixed carbon: Fixed carbon was calculated by difference using the data obtained above for moisture content, volatile matter, and ash content.

#### 3.3.2. Ultimate Analysis

Basic elemental composition: The amount of carbon (C), hydrogen (H), nitrogen (N) in samples were measured in an Elemental Analysis Unit FISONS Carlo Erba EA1108 (FISONS, Milan, Italy) with a CHNS Microanalyzer. The oxygen (O) content was determined by subtracting the sum of the other element contents from 100%.

Higher heating value: The higher heating values (HHV) of samples were calculated through a simple Equation (1) based on data obtained from proximate analysis as proposed by Cordero [27], in which volatile matter (VM) and ash are used, and the results were reported as MJ/kg.
(1)$$\mathrm{HHV}\;=\frac{35,430-183.5\times \mathrm{VM}-354.3\times \mathrm{ASH}}{1000}$$

Thermal analysis: The thermochemical characterization of seaweeds was studied through thermogravimetric (TG), derivative thermogravimetric analyses (DTG), and differential scanning calorimetry (DSC) in the simultaneous thermal analyzer SETSYS Evolution (Setaram Instrumentation, Caluire-et-Cuire, France). Non-isothermal assays were performed by heating samples (7–15 mg) at ramp rates of 10 °C/min, in the temperatures range of 20–900 °C using nitrogen as carrier gas (30 mL/min). Then, isothermal experiments were performed at 900 °C under oxidizing air atmosphere (30 mL/min). The sample weight and furnace temperature were registered during thermal analysis. TGA, DTG, and DSC thermograms were plotted that show the mass loss (%), the mass loss rate (%/min), and heat flow (mW), respectively, during thermal analysis.

### 3.4. Evaluation of Seaweeds as a Promising Source of Bioactive Compounds

Total phenolic content (TPC), total carotenoid content (TCC), antioxidant, anti-inflammatory, and cytotoxic activities were found on aqueous–organic extracts of the studied seaweeds.

Heat-assisted extraction was performed according to the methodology described by Perez-Jimenez [34] with some modifications. Samples (1 g) were placed in amber glass bottles with 20 mL of acidified aqueous methanol (50% *v*/*v*, pH = 2) (Carlo Erba Reagents, Milan, Italy) to obtain a solid/liquid ratio of 50 g/L. The bottles holding the mixture were placed in a thermostatic water bath at 40 °C and stirred using a magnetic stirrer for 1 h. Then, the resulting mixtures were transferred to Falcon tubes to be centrifugated at 6000 rpm for 15 min. The supernatants were recovered, and 20 mL of aqueous acetone (70% *v*/*v*) (Carlo Erba Reagents, Milan, Italy) was added to residues. The new mixtures were subjected to the same extraction procedure stated before. Then, methanolic and acetonic extracts were combined and used to find the extractable bioactive compounds described above and their associated bioactivities.

The extraction yield was thermogravimetrically determined according to the methodology explained by Silva [33] and was calculated using Equation (2) and expressed as g extract/100 g dry weight. Determinations were conducted in duplicate.
(2)$$\mathrm{Yield}\;\left(\%\right)=\frac{P_{t=24\;h}-P_{t=0}}{\left(\frac{m_{\mathrm{sw}}\times V_{a}}{V_{\mathrm{sv}}}\right)\times \left(\frac{100-{\mathrm{MC}}_{\mathrm{sw}}}{100}\right)}\times 100$$
where,

P_t=0_: mass of crucible before adding the extracted solutionP_t=24 h_: mass of crucible after 24 h of dryingm_sw_: mass of dry seaweedV_a_: volume of extracted solution aliquot (5 mL)V_sv_: volume of solvents used for extraction (40 mL)MC_sw_: moisture content (%) of each seaweed

#### 3.4.1. Total Phenolic Content

TPC was spectrophotometrically determined using the Folin–Ciocalteu reagent (FCR) (VWR Chemicals, Fontenary-Sous-Bois, France) according to the methodology described by Cassani [50]. Briefly, 25 µL of each extract (serially diluted with water) was added to 125 µL of FCR (diluted in water, 1:10). After 3 min of incubation at room temperature, 100 µL of Na_2_CO_3_ solution (7.5% *w*/*v*) (Carlo Erba Reagents, Milan, Italy) was added and the reaction mixture was incubated for 2 h at the same temperature. Absorbance was measured at 765 nm using a Synergy HTX multi-mode reader (Bio-Tek, Winooski, VT, USA). TPC was calculated using gallic acid as standard. The calibration curve of gallic acid was made up in the range of 5–100 µg/mL. Results were reported as µg gallic acid (Merck, Darmstad, Germany) equivalents (GAE)/g DM. Determinations were conducted in duplicate.

#### 3.4.2. Total Carotenoid Content

TCC was spectrophotometrically found and thus, the absorbance of extracted solutions was measured at 450 nm using a Synergy HTX multi-mode reader (Bio-Tek, Winooski, VT, USA). TCC was calculated according to Equation (3) proposed by Scott [51]. Results were reported as µg total carotenoids/g DM. Determinations were conducted in duplicate.
(3)$$\mathrm{TCC}\;=\frac{A_{450}}{2500}\times 10\frac{\mathrm{mg}}{\mathrm{mL}}\times \frac{V_{\mathrm{sv}}}{m_{\mathrm{sw}}\times \frac{100-{\mathrm{MC}}_{\mathrm{sw}}}{100}}$$
where,

A_450_: absorbance measured at 450 nmm_sw_: mass of dry seaweedV_sv_: volume of solvents used for extraction (40 mL)MC_sw_: moisture content (%) of each seaweed

#### 3.4.3. Antioxidant Activity

Total antioxidant activity of seaweeds was studied through four assays: the scavenging activity of the stable 1,1-diphenyl-2-picrylhydrazyl (DPPH) radical, the Trolox equivalents antioxidant capacity (TEAC), β-carotene, and crocin bleaching antioxidant assays.

DPPH assay

A stock methanolic solution (7.61 mM) of DPPH radical (Alfa Aesar, Thermo Fisher, Kandel, Germany) was prepared. Then, the stock reagent was diluted 1:50 with methanol to obtain 1.4 units of absorbance measured at 515 nm. The reaction was conducted in 96-well microplates, where 50 µL of each extract (serially diluted with methanol) was mixed with 200 µL of the DPPH stock solution. The resulting mixture was incubated at room temperature in the dark for 60 min. Absorbance was measured at 515 nm at once after mixing (t = 0) and after 60 min using a Synergy HTX multi-mode reader (Bio-Tek, Winooski, VT, USA). A control sample was also prepared by replacing the sample with solvent. The scavenging activity of seaweed extracts was calculated following Equation (4).
(4)$$\mathrm{DPPH}\;\left(\%\right)=\left(\frac{A_{c}-A_{s}}{A_{c}}\right)\times 100$$
where,

A_c_: absorbance of control at t = 0A_s_: absorbance of sample at t = 60 min

DPPH scavenging activity (%) vs. extract concentration (mg/mL) were plotted for all seaweeds to model data. Linear and non-linear methods (Equations (5) and (6), respectively) were used to adjust experimental data.
(5)Y = a×X+b
(6)Y = a×ln(X)+b
where, *a* and *b* are the model parameters.

From these studies, the model parameters and determination coefficients (R^2^) were obtained for each seaweed and the effective concentration value (*IC*_50_), which refers to the extract concentration needed to scavenge 50% of DPPH radicals, was determined.

TEAC assay

This method is based on the antioxidant compounds’ ability to scavenge the ABTS^•+^ radical, compared with Trolox’s scavenging ability, an analog of vitamin E. This method was carried out following the methodology proposed by Cassani [35]. In this regard, the ABTS^•+^ radical was produced by the interaction of ABTS (19.3 mg) (Alfa Aesar, Thermo Fisher, Kandel, Germany) dissolved in 5 mL of distilled water and 88 μL of potassium persulfate (K_2_S_2_O_8_, 0.0378 g/mL) (Carlo Erba Reagents, Milan, Italy). The radical solution was incubated at room temperature for 16 h. Then, 250 μL of the activated ABTS^•+^ radical was mixed with 10 mL of ethanol (Carlo Erba Reagents, Milan, Italy). Under these conditions, the absorbance of the reagent solution is ∼1.4 measured at 734 nm. The reaction was produced by adding 200 µL of the previously prepared ABTS^•+^ radical solution and 50 µL of extract solutions (serially diluted with ethanol) in 96-well microplates. Immediately after mixing (t = 0) and after 60 min of incubation at room temperature, absorbance at 734 nm was measured using a Synergy HTX multi-mode reader (Bio-Tek, Winooski, VT, USA). A control sample was also prepared by replacing the sample with solvent. The ABTS scavenging activity of seaweed extracts was also calculated according to Equation (4) and linear and non-linear methods (Equations (5) and (6)) were also used to adjust experimental data derived from the plot of ABTS scavenging activity (%) vs. extract concentration (mg/mL). Similar to the DPPH assay, the model parameters and determination coefficients (R^2^) were obtained for each seaweed and the effective concentration value (*IC*_50_), which refers to the extract concentration needed to scavenge 50% of ABTS radicals, was determined.

β-carotene bleaching assay

This method is based on the ability of antioxidant compounds present in seaweed extracts to delay the oxidative discoloration of β-carotene in a β-carotene/linoleic acid emulsion. Linoleic acid is easily oxidized at 50 °C and the radicals formed may react with the unsaturated β-carotene chain causing its oxidation with the corresponding decrease in absorbance at 470 nm. In presence of an antioxidant compound source, β-carotene oxidation is delayed since antioxidants may neutralize the linoleate free radical and other free radicals formed in the oxidation reaction [32]. Thus, this method is suitable for studying the capacity of lipophilic antioxidants to protect the β-carotene molecule and inhibit or delay lipid oxidation [52].

For this purpose, a stock solution of β-carotene (Tokyo Chemical Industry C.O., Tokyo, Japan) was prepared by mixing β-carotene (4 mg), Tween-40 (4 g) (Carlo Erba Reagents, Milan, Italy), linoleic acid (0.5 mL) (Acros organics, Morris Planes, NJ, USA) and chloroform (20 mL) (Carlo Erba Reagents, Milan, Italy) in a round-bottomed flask following the methodology proposed by Prieto [53]. Then, chloroform was completely recovered using a rotary evaporator at 45 °C for 15 min. An aliquot (0.25 mL) of the resulting oily residue was vigorously shaken with 30 mL of preheated Milli-Q water (45 °C) having 100 mM of phosphate buffer (pH = 6.5). Under these conditions, the absorbance of the β-carotene solution is ∼1.4, measured at 470 nm. The reaction mixture was produced by adding 200 µL of the previously prepared β-carotene solution and 50 µL of sample (serially diluted with solvent) in 96-well microplates. The absorbance of the final reaction was at once determined after mixing (t = 0) at 470 nm using a Synergy HTX multi-mode reader (Bio-Tek, Winooski, VT, USA). Then, microplates were incubated at 45 °C under continuous shaking (85 rpm) and successive absorbance measurements were conducted at regular intervals (20 min) for 200 min. A control sample was also prepared, by replacing the sample with solvent [53]. For quantification, the time course of the oxidative response was defined according to Equation (7).
(7)$$R=1-\frac{S_{t}}{S_{0}}$$
where, S_0_ and S_t_ are the substrate (β-carotene) concentrations at times 0 and t, respectively.

Then, the oxidative response (Equation (7)) was adjusted to a kinetic profile based on the Weibull mass function, Equation (8), according to the method explained by Prieto [53].
(8)$$R=K\times \left(1-e^{-\left(ln2\right)\times {\left(\frac{t}{\tau }\right)}^{\alpha }}\right)$$
where,

K: asymptoteτ: time when 50% oxidation is achievedα: shape parameter associated with the maximum slope of the response (v_max_)

The parameters of Weibull models were fitted using the non-linear least-squares (quasi-Newton) method provided by the macro-Solver of the Microsoft Excel 2016 spreadsheet. Determinations were conducted in duplicate and results were expressed as the time (min) when 50% oxidation was achieved per mg of extract.

Crocin bleaching assay

The basis of this method is like that previously explained for the β-carotene assay. In this sense, 2,2′-Azobis(2-amidinopropane) dihydrochloride (AAPH) is used as a free radical source that can induce crocin oxidation (substrate) with the corresponding decrease in absorbance at 450 nm. In the presence of antioxidants, crocin oxidation is delayed since antioxidants react with the formed radicals. This method is suitable to analyze the performance of hydrophilic antioxidants on protecting crocin against free radicals.

To carry this out, crocin (100 μmol/L) (Tokyo Chemical Industry C.O., Tokyo, Japan) and AAPH (7.68 mmol/L) (Acros organics, Morris Planes, NJ, USA) solution were dissolved and mixed in 30 mL of preheated Milli-Q water (40 °C) containing 100 mM of phosphate buffer (pH = 5.5), according to the methodology proposed by Prieto [54]. It is worth mentioning that the resulting mixture should be prepared just before use to avoid the degradation of any reagent. Under these conditions, the absorbance of the reagent solution is ∼1.4, measured at 450 nm. The reaction mixture was produced by adding 200 µL of the previously prepared reagents solution and 50 µL of sample (serially diluted with solvent) in 96-well microplates. The absorbance of the final reaction was at once determined after mixing (t = 0) at 450 nm using a Synergy HTX multi-mode reader (Bio-Tek, Winooski, VT, USA). Then, microplates were incubated at 37 °C under continuous shaking (85 rpm) and successive measurements were conducted at regular intervals (20 min) for 120 min. A control sample was also made by replacing the sample with solvent. For quantification, data were fitted to the Weibull method as previously described, and results were expressed as the time (min) when 50% oxidation was achieved per mg of extract. Determinations were conducted in duplicate.

#### 3.4.4. Anti-Inflammatory Activity

The anti-inflammatory activity of seaweed extracts was determined as a function of the inhibition of nitric oxide (NO) production by lipopolysaccharide (LPS)-induced RAW264.7 macrophages cell line, as described by Garcia-Perez [44]. For the purpose, extracts were lyophilized to obtain the dry residue, which was resuspended in water and assessed at four concentrations, ranging 0.125–0.4 mg/mL. The murine macrophage cell line was cultured in DMEM medium (Dulbecco’s Modified Eagle Medium, Gibco, UK) supplemented with 10% heat-inactivated fetal bovine serum (Fisher Scientific, Loughborough, UK), 1% L-glutamine (200 mM) (Gibco, Fisher Scientific, Madrid, Spain), 1% MEM essential amino acid solution (100X) (Gibco, UK), and 1% antibiotic solution (Fisher Scientific, Rochester, NY, USA) holding 50 U/mL streptomycin and 100 mg/L streptomycin. Cells were kept at 37 °C under a humidified 5% CO_2_ atmosphere, and cell density was adjusted at 5 × 10^5^ cells/mL after assessing cell viability was >99% according to Trypan blue exclusion assay. The cell suspension was later seeded in 96-well microplates to reach an experimental cell density of 1.5 × 10^5^ cells/well and incubated in the same conditions for 24 h. Afterward, cells were treated with algal extracts and incubated for 1 h, and further stimulated by LPS (Enzo Life Sciences Inc., New York, NY, USA), which was added at 1 mg/mL, and incubated for 18 h. Dexamethasone was used as a positive control. In all cases, dexamethasone, the extracts, and LPS were diluted in supplemented DMEM medium. The NO determination was found in the cell supernatants using the Griess reagent system kit (Promega, Madison, WI, USA), following the manufacturers’ instructions. All determinations were conducted in triplicate and results were expressed as effective concentration (*IC*_50_), which stands for the extract concentration required to inhibit by 50% the NO production caused by LPS induction, in µg/mL.

#### 3.4.5. Cytotoxic Activity

Three cancer cell lines and one non-tumor cell line were employed to assess the cytotoxic activity of seaweed extracts, applying the sulforhodamin B (SRB) cytotoxicity assay, namely: the African green monkey kidney-derived Vero cell line, the human gastric cancer AGS cell line, the human lung adenocarcinoma A549 cell line, and the human hepatocarcinoma HepG2 cell line. Vero, A549, and HepG2 were routinely kept in DMEM medium, while AGS was cultured in RPMI-1640 medium, both supplemented as stated above and incubated as previously described. A cell suspension with 5 × 10^4^ cells/mL was obtained and the viability was assessed at >99% using Trypan blue exclusion assay and plated in a 96-well microplate with different concentrations of algal extracts, ranging 0.125–0.4 mg/mL, for 48 h under the same conditions. Ellipticine (Enzo, USA) was used as a positive control. After incubation, 100 µL of cold 10% trichloroacetic acid (Fisher Chemical Reagents, Pittsburgh , USA) were added and the mixture was incubated for 1 h at 4 °C to allow cell attachment. Afterward, the mixture was removed and 100 µL of 0.057% (*w*/*v*) SRB (Alfa Aesar, Thermo Fisher, Germany) prepared in 1% acetic acid, and samples were incubated at room temperature for 30 min to allow cell staining. Finally, the SRB excess was removed washing three times with 1% acetic acid (Fisher Chemical Reagents, Pittsburgh , USA), and the remaining SRB was redissolved in 200 µL of 10 mM Tris solution for 30 min in an orbital shaker at 250 rpm. The absorbance of samples was measured at 540 nm in a Synergy HTX multi-mode reader (Bio-Tek, Winooski, VT, USA). All determinations were conducted in triplicate and results were expressed as growth inhibitory 50 (*GI*_50_) concentration, which stands for the extract concentration needed to inhibit cell growth by 50%, in µg/mL.

### 3.5. Statistical Analysis

Three independent experimental runs with each experiment performed at least in duplicate were carried out. Results were reported as mean values ± standard deviation (SD). R software version 2.12 (R Development Core Team, 2011) was used to analyze data. Analysis of variance (ANOVA one-way, *p* < 0.05) and Tukey–Kramer comparison test were performed to study the matrix effect on extraction yield, TPC, TCC, antioxidant, anti-inflammatory, and cytotoxic activities of samples.

## 4. Conclusions

The present work shows that some seaweed species have a high content of polysaccharides (including total fiber), low fat content, moderate protein content, a high proportion of ω3 fatty acids, which favors a healthy ω6:ω3 ratio, and high content of essential minerals. This supports their use as food or functional ingredients to improve certain nutritional properties in food formulations, as well as for the extraction of their biological compounds.

Concerning thermal characterization, three biomass decomposition stages were identified with the maximum degradation rate recorded at 250 °C. These values were lower than those reported for terrestrial energy crops, due to different polysaccharide compositions, in particular the lack of hemicellulose and lignin in the carbohydrate profile. Macroalgae species with the lowest ash content and highest volatile matter and fixed carbon could be promising resources for alternative biomass-based renewable fuel production with the added advantages of having higher growth rates, no requirements of arable land for cultivation, and higher carbon dioxide mitigation ability over terrestrial biomasses.

Regarding bioactive compounds, high extraction yield for most of the seaweeds was recorded suggesting a higher release of polar soluble compounds such as phytochemical compounds, polysaccharides, proteins, peptides, and organic acids from these species. Different behaviors of seaweed extracts were seen between the four antioxidant activity assays studied. The complex phytochemical structures present in seaweed extracts may have different solubility in each solvent used in the antioxidant activity assays. It is noteworthy that *H. elongata* showed the highest phytochemical content and antioxidant activities among the studied species. By contrast, *P. purpurea* showed moderate performance on anti-inflammatory activity.

## Figures and Tables

**Figure 1 ijms-23-02355-f001:**
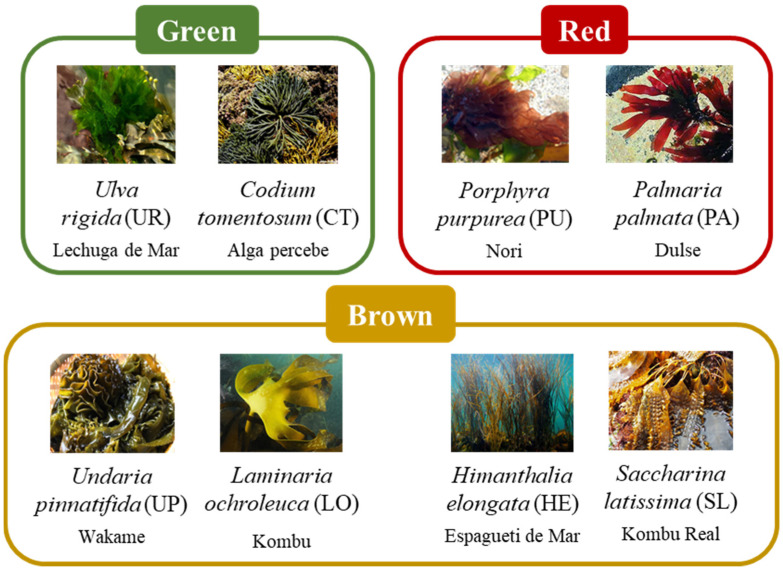
Seaweeds included in this study with their Spanish common names.

**Figure 2 ijms-23-02355-f002:**
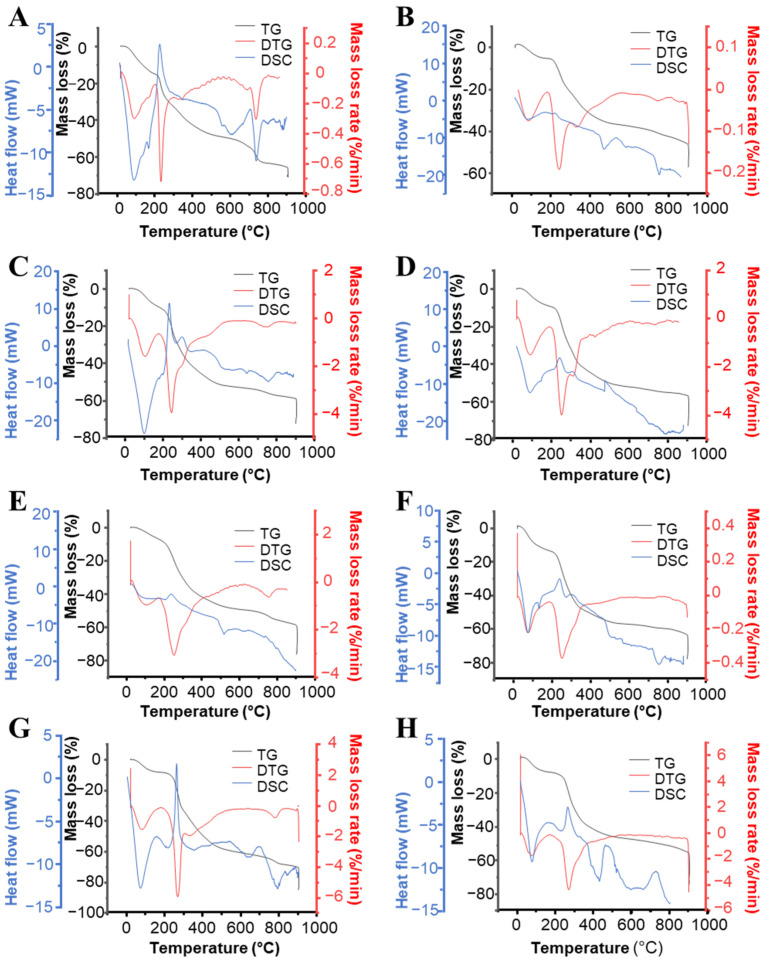
TG, DTG, and DSC curves of the studied seaweeds. Different figures show the studied algae: Part (**A**), *Ulva rigida*; (**B**), *Codium tomentosum*; (**C**), *Undaria pinnatifida*; (**D**), *Laminaria ochroleuca*; (**E**), *Himanthalia elongata*; (**F**), *Saccharina latissima*; (**G**), *Porphyra purpurea*; (**H**), *Palmaria palmata*.

**Table 1 ijms-23-02355-t001:** Nutritional composition of the studied seaweeds (g/100 g dry weight).

Seaweed	Nutritional Composition		
	Lipids	Protein	Carbohydrates
*UR*	0.50	18.45	48.57
*CT*	0.61	16.36	41.54
*UP*	1.24	16.49	52.53
*LO*	0.72	10.64	59.09
*HE*	1.74	9.81	64.30
*SL*	0.33	10.34	65.92
*PU*	0.61	34.79	48.58
*PA*	4.22	23.55	50.41

*UR*, *Ulva rigida*; *CT*, *Codium tomentosum*; *UP*, *Undaria pinnatifida*; *LO*, *Laminaria ochroleuca*; *HE*, *Himanthalia elongata*; *SL*, *Saccharina latissima*; *PU*, *Porphyra purpurea*; *PA*, *Palmaria palmata*.

**Table 2 ijms-23-02355-t002:** Fatty acid content of the studied seaweeds (% of total fatty acid content).

Fatty Acid		Seaweed							
		*UR*	*CT*	*UP*	*LO*	*HE*	*SL*	*PU*	*PA*
C12:0 (Lauric acid)		*nd*	4.93	*nd*	*nd*	*nd*	1.98	*nd*	*nd*
C13:0 (Tridecanoic acid)		*nd*	*nd*	0.89	*nd*	0.48	0.79	*nd*	*nd*
C14:0 (Myristic acid)		*nd*	2.49	3.97	8.59	9.55	6.62	*nd*	5.74
C15:0 (Pentadecanoic acid)		1.35	0.63	0.89	0.90	0.66	1.05	*nd*	*nd*
C16:0 (Palmitic acid)		21.07	32.79	22.78	32.58	31.75	37.53	35.19	42.05
C17:0 (Heptadecanoic acid)		*nd*	0.22	*nd*	*nd*	0.32	0.90	*nd*	*nd*
C18:0 (Stearic acid)		6.74	1.92	3.17	1.72	2.02	5.73	8.76	16.33
C20:0 (Arachidic acid)		*nd*	1.46	*nd*	0.61	0.72	1.90	*nd*	*nd*
C21:0 (Heneicosanoic acid)		*nd*	1.70	*nd*	*nd*	*nd*	*nd*	*nd*	*nd*
C22:0 (Behenic acid)		2.68	8.79	*nd*	*nd*	0.88	*nd*	*nd*	*nd*
C24:0 (Lignoceric acid)		*nd*	2.72	*nd*	*nd*	0.63	*nd*	*nd*	*nd*
**Sum SATURATED FAs**		31.84	57.63	31.70	44.39	47.01	56.49	43.95	64.12
C14:1 (Myristoleic acid)		*nd*	*nd*	*nd*	0.88	0.27	*nd*	*nd*	*nd*
C15:1 (cis-10-Pentadecenoic acid)		8.88	10.53	9.65	5.70	5.53	1.54	8.55	5.03
C16:1 (Palmitoleic acid)		1.65	2.77	1.05	6.63	2.16	5.07	*nd*	4.63
C17:1 (cis-10-Heptadecanoic acid)		1.21	*nd*	*nd*	*nd*	*nd*	0.91	*nd*	*nd*
C18:1 trans (trans-9-Elaidic acid)		*nd*	*nd*	0.26	*nd*	*nd*	0.77	*nd*	*nd*
C18:1 cis (Oleic acid)		29.51	14.02	12.41	18.77	15.03	21.23	17.13	9.62
C20:1 (cis-11-Eicosenoic acid)		*nd*	0.36	*nd*	*nd*	*nd*	*nd*	3.43	*nd*
C24:1 (Nervonic acid)		*nd*	*nd*	*nd*	*nd*	*nd*	*nd*	*nd*	1.59
**Sum MONOUNSATURATED FAs (MUFAs)**		41.26	27.67	23.38	31.97	22.99	29.51	29.11	20.87
C18:2 trans (Linolelaidic acid)		*nd*	*nd*	*nd*	*nd*	0.12	*nd*	*nd*	*nd*
C18:2 cis (Linoleic acid)		9.62	4.46	16.08	12.61	14.27	8.08	13.02	3.71
C18:3 ω6 (γ-Linolenic acid)		0.99	0.57	2.97	1.44	1.45	0.87	*nd*	*nd*
C18:3 ω3 (Linolenic acid)		14.25	8.53	24.72	8.09	11.46	2.79	5.24	5.61
C20:2 (cis-11,14-Eicosadienoic acid)		*nd*	0.60	*nd*	0.50	0.22	*nd*	1.90	*nd*
C20:3 ω6 (cis-8,11,14-Eicosatrienoic acid)		*nd*	0.27	1.15	0.64	2.30	0.84	3.62	*nd*
C20:4 (cis-5,8,11,14-Eicosatetraenoic acid)		2.04	0.25	*nd*	0.35	0.17	1.41	3.16	3.53
C22:6 ω3 (cis-4,7,10,13,16,19-Docosahexaenoic acid)		*nd*	*nd*	*nd*	*nd*	*nd*	*nd*	*nd*	2.17
**Sum POLYUNSATURATED FAs (PUFAs)**		26.90	14.70	44.92	23.63	30.00	13.99	26.94	15.02
	Sum PUFAs ω6	10.61	5.30	20.19	14.69	18.15	9.79	16.64	3.71
	Sum PUFAs ω3	14.25	8.53	24.72	8.09	11.46	2.79	5.24	7.78
	Ratio ω6/ω3	0.74	0.62	0.82	1.81	1.58	3.50	3.18	0.48

*nd*, *not detected*; *UR*, *Ulva rigida*; *CT*, *Codium tomentosum*; *UP*, *Undaria pinnatifida*; *LO*, *Laminaria ochroleuca*; *HE*, *Himanthalia elongata*; *SL*, *Saccharina latissima*; *PU*, *Porphyra purpurea*; *PA*, *Palmaria palmata*.

**Table 3 ijms-23-02355-t003:** Composition of elements in the studied seaweeds (mg/kg DM).

Seaweed	Microelements								Macroelements			
	Fe	Mn	Cr	Mo	Cu	Zn	Se	I	Ca	K	Mg	P
*UR*	1177	80.1	2.5	<2	4.12	6	<4.8	138	2778	19,792	22,932	2018
*CT*	125	32.2	<2	<2	<2	<1	<4.8	224	5139	6794	10,426	1372
*UP*	62	8.1	<2	<2	<2	28	<4.8	154	6485	65,485	4106	6604
*LO*	28	3.7	<2	<2	<2	4	<4.8	5829	8077	151,874	4660	1537
*HE*	16	17.2	<2	<2	<2	25	<4.8	189	9224	73,477	8015	1393
*SL*	31	3.0	<2	<2	<2	15	<4.8	2797	8704	47,741	7162	1657
*PU*	246	28.1	<2	<2	7.90	33	<4.8	82	1494	33,556	3109	5867
*PA*	177	19.7	<2	<2	3.92	21	<4.8	386	4756	100,836	3870	3392

*UR*, *Ulva rigida*; *CT*, *Codium tomentosum*; *UP*, *Undaria pinnatifida*; *LO*, *Laminaria ochroleuca*; *HE*, *Himanthalia elongata*; *SL*, *Saccharina latissima*; *PU*, *Porphyra purpurea*; *PA*, *Palmaria palmata*.

**Table 4 ijms-23-02355-t004:** Proximate and ultimate analysis of the studied seaweeds.

Seaweed	Proximate Analysis				Ultimate Analysis				
	Moisture	Ash	Volatile	Fixed C	N	C	H	O	HHV
	(%)	(%)	(%)	(%)	(%)	(%)	(%)	(%)	(MJ/kg)
*UR*	11.7	28.7	53.8	5.8	2.6	25.3	4.6	67.5	15.39
*CT*	5.9	38.9	41.3	13.9	2.5	23.1	4.4	70.0	14.07
*UP*	9.9	27.4	49.4	13.3	2.4	24.0	3.9	69.7	16.66
*LO*	9.6	26.9	47.8	15.7	1.5	25.2	4.4	68.8	17.13
*HE*	6.8	23.6	51.8	17.8	1.5	27.3	4.7	66.5	17.56
*SL*	14.9	20.1	50.3	14.7	1.4	27.3	5.3	66.0	19.08
*PU*	8.5	15.1	61.8	14.6	5.1	37.5	6.9	50.6	18.74
*PA*	8.7	23.8	48.4	19.1	3.4	29.5	5.2	61.8	18.12

HHV, Higher heating value determined according to the equation proposed by Cordero [27]; *UR*, *Ulva rigida*; *CT*, *Codium tomentosum*; *UP*, *Undaria pinnatifida*; *LO*, *Laminaria ochroleuca*; *HE*, *Himanthalia elongata*; *SL*, *Saccharina latissima*; *PU*, *Porphyra purpurea*; *PA*, *Palmaria palmata*.

**Table 5 ijms-23-02355-t005:** Bioactive compound composition and antioxidant activity of the studied seaweeds.

Seaweed	Yield	TPC	TCC	Antioxidant Activity			
				DPPH	ABTS	βC	CROCIN
	(%)	(µg GAE/g dw)	(µg C/g dw)	*IC*_50_ (mg/mL)	*IC*_50_ (mg/mL)	(min/mg E)	(min/mg E)
*UR*	26.58 ± 1.98 ^d^	366.48 ± 56.02 ^d^	282.61 ± 54.10 ^a^	n.a	n.a	761.69	69.18
*CT*	44.30 ± 1.68 ^ab^	916.62 ± 121.04 ^bc^	295.24 ± 14.80 ^a^	n.a	7.73 ± 0.10 ^g^	190.23	59.36
*UP*	41.24 ± 2.87 ^bc^	682.56 ± 51.15 ^cd^	184.85 ± 20.66 ^bc^	n.a	2.01 ± 0.02 ^e^	916.91	43.98
*LO*	44.59 ± 1.66 ^ab^	527.99 ± 83.91 ^cd^	229.42 ± 21.10 ^abc^	n.a	0.50 ± 0.00 ^a^	266.85	43.34
*HE*	38.45 ± 2.57 ^c^	3044.29 ± 387.20 ^a^	134.33 ± 14.06 ^cd^	5.78 ± 0.26	0.70 ± 0.03 ^b^	1678.50	143.94
*SL*	47.29 ± 0.24 ^a^	1207.10 ± 131.61 ^b^	75.76 ± 3.43 ^d^	n.a	1.38 ± 0.00 ^c^	181.28	56.38
*PU*	20.55 ± 0.51 ^e^	3448.55 ± 241.33 ^a^	227.68 ± 36.95 ^abc^	n.a	3.04 ± 0.05 ^f^	2401.40	210.86
*PA*	37.28 ± 0.58 ^c^	931.81 ± 241.12 ^bc^	61.27 ± 0.04 ^d^	n.a	1.81 ± 0.01 ^d^	414.16	38.36

TPC, Total phenolic content; TCC, total carotenoid content; GAE, gallic acid equivalents; C, carotenoids; *IC*_50_, extract concentration which is required to scavenge 50% of the DPPH/ABTS radicals. E, algae extract. *UR*, *Ulva rigida*; *CT*, *Codium tomentosum*; *UP*, *Undaria pinnatifida*; *LO*, *Laminaria ochroleuca*; *HE*, *Himanthalia elongata*; *SL*, *Saccharina latissima*; *PU*, *Porphyra purpurea*; *PA*, *Palmaria palmata*. Different letters represent significant differences.

**Table 6 ijms-23-02355-t006:** Anti-inflammatory and cytotoxic activity of algal extracts.

Cell Lines	Seaweed								Control	
	*UR*	*CT*	*UP*	*LO*	*HE*	*SL*	*PU*	*PA*	*Dexa*	*Ellip*
**Anti-inflammatory activity (*IC*_50_, µg/mL)**										
RAW264.7	341.8 ± 20.2 ^b^	264.1 ± 5.6 ^a^	>400	>400	>400	>400	193.1 ± 4.5 ^a^	368.7 ± 30.7 ^b^	7.2 ± 0.9	-
**Cytotoxic activity (*GI*_50_, µg/mL)**										
Vero	>400	>400	>400	>400	>400	>400	>400	>400	-	0.35 ± 0.08
AGS	>400	>400	>400	>400	>400	>400	>400	>400	-	0.33 ± 0.06
A549	>400	>400	>400	>400	>400	>400	>400	>400	-	0.95 ± 0.03
HepG2	>400	>400	>400	>400	>400	>400	379.2 ± 27.2	>400	-	0.23 ± 0.03

*UR*, *Ulva rigida*; *CT*, *Codium tomentosum*; *UP*, *Undaria pinnatifida*; *LO*, *Laminaria ochroleuca*; *HE*, *Himanthalia elongata*; *SL*, *Saccharina latissima*; *PU*, *Porphyra purpurea*; *PA*, *Palmaria palmata*. Vero, the African green monkey kidney-derived cell line; AGS, the human gastric cancer cell line; A549, the human lung adenocarcinoma cell line; HepG2, the human hepatocarcinoma cell line. Different letters represent significant differences.

## Data Availability

Not applicable.

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
