# Peer review of "Thermochemical Characterization of Eight Seaweed Species and Evaluation of Their Potential Use as an Alternative for Biofuel Production and Source of Bioactive Compounds"

_ijms, 2022, doi:10.3390/ijms23042355_

Round 1

Reviewer 1 Report

The quest for improved organisms for the benefit of humankind are essential to better utilize resources and improve human health. As such, the manuscript performed a varied analysis of 8 various seaweed species for their potential industrial exploitation. Overall the manuscript is very well written, clear and the authors have provided an exhaustive analysis of these organisms. I recommend the manuscript for publication after correction of a few minor comments below.

Line 59. Please provide references for this statement.

Line 65. since ‘it’ has….

Figure 2 is very poor quality and needs to be fixed.  The writing around all axis should be legible and the box for each graph could be made larger.

Perhaps the data tables (especially those with statistical analysis) could be represented in a graphical format for example a bar graph or something similar. This would really help in comparing the values across the different samples.

Author Response

Dear reviewer, 

Thank you for your important comments on the improvement of the manuscript. We want to apologize for the delay in responding to the comments. Hope this new version has the expected quality. 

Please, find next the responses to your comments. We deeply revised the manuscript and responded to each issue raised by each of the reviewers in ‘red color’.

Changes in the revised version of the manuscript including text, figures and tables were done with the Track & Changes tool and newly added references were highlighted in yellow.

Yours sincerely,

The authors

The quest for improved organisms for the benefit of humankind are essential to better utilize resources and improve human health. As such, the manuscript performed a varied analysis of 8 various seaweed species for their potential industrial exploitation. Overall the manuscript is very well written, clear and the authors have provided an exhaustive analysis of these organisms. I recommend the manuscript for publication after correction of a few minor comments below.

We are very grateful for the reviewer’s observation.

1 - Line 59. Please provide references for this statement.

Reference was added for that sentence (page 2, line 60).

2 - Line 65. since ‘it’ has….

We appreciate the reviewer’s observation, and it was corrected.

3 - Figure 2 is very poor quality and needs to be fixed. The writing around all axis should be legible and the box for each graph could be made larger.

Figure 2 was improved according to the reviewer’s suggestion.

4 - Perhaps the data tables (especially those with statistical analysis) could be represented in a graphical format for example a bar graph or something similar. This would really help in comparing the values across the different samples.

We are grateful for the reviewer’s observation, but Table 5 was kept in its original version because we consider that data is well-presented and summarized in such a format. However, to facilitate analysis, data regarding extraction yield, total phenolic content, and total carotenoid content of the studied seaweeds have also been represented in a graphical format as supplementary material (Figure S1).

Reviewer 2 Report

This paper is interesting and definitely worth publishing, but a few aspects should be reviewed and corrected or completed, as follows:

Lines 56-62: these statements have to be supported by appropriate references.

Lines 240, 246, 265-266: the model and commercial source of the microplate reader should be provided. The sources of reagents and microplates used should also be stated.

Lines 242-243, 248, 348: triplicate should have been preferrable to duplicate measurements.

Lines 288-289: because the graphical method is not very accurate, estimating IC50 through non-linear regression is currently the de facto practice in the field.

Line 352: after “as described elsewhere”, an appropriate reference should be provided.

Lines 354, 380: the concentrations used were relatively high (in the mM range), and lower (n the nano- or uM range) would have been preferrable.

Line 378: cell density to be corrected to “5 × 10^4 cells/mL”

Lines 419-421, 424-428: these statements need appropriate referencing.

Line 541: because enzymes are proteins, they do not need separate mentioning.

Author Response

Dear reviewer,

Thank you for your important comments on the improvement of the manuscript. We want to apologize for the delay in responding to the comments. Hope this new version has the expected quality.

Please, find next the responses to your comments. We deeply revised the manuscript and responded to each issue raised by each of the reviewers in ‘red color’.

Changes in the revised version of the manuscript including text, figures and tables were done with the Track & Changes tool and newly added references were highlighted in yellow.

Yours sincerely,

The authors

This paper is interesting and definitely worth publishing, but a few aspects should be reviewed and corrected or completed, as follows:

We are very grateful for the reviewer’s constructive comments.

1 - Lines 56-62: these statements have to be supported by appropriate references.

References supporting the statements made on page 2, lines 60 and 64 were added.

2 - Lines 240, 246, 265-266: the model and commercial source of the microplate reader should be provided. The sources of reagents and microplates used should also be stated.

Information regarding the commercial source of microplate reader and reagents used for bioactive compounds determination was added.

3 - Lines 242-243, 248, 348: triplicate should have been preferrable to duplicate measurements.

We appreciate the reviewer’s observation. In this study, a completely randomized experimental design of three independent experimental runs with each experiment performed at least in duplicate was carried out. This was clarified in the revised version of the manuscript (page 10, lines 418-419).

4 - Lines 288-289: because the graphical method is not very accurate, estimating IC50 through non-linear regression is currently the de facto practice in the field.

Linear non-linear regression methods were used to adjust experimental data. From these analyses, the model parameters, and determination coefficients (R2) were obtained. An additional table was included in supplementary material showing the statistical parameters obtained from models. This was clarified in the revised version of manuscript.

5 - Line 352: after “as described elsewhere”, an appropriate reference should be provided.

The reviewer is right, a reference was missing at the end of the sentence. A recent appropriate reference was added accordingly (page 9, line 375).

6 - Lines 354, 380: the concentrations used were relatively high (in the mM range), and lower (n the nano- or uM range) would have been preferrable.

The reviewer is right. As an initial screening of raw extracts, a wide range of concentrations was tested for the determination of both anti-inflammatory and cytotoxic activities from 125 to 400 µM. However, due to the moderate activity determined (>400 micrograms per liter), the use of lower concentrations would not make any difference at the definitive results, as even higher extract concentrations would be required to achieve IC50 values for most cases.

7 - Line 378: cell density to be corrected to “5 × 10^4 cells/mL”

We appreciate the reviewer’s observation, and the unit format was corrected (page 10, line 402).

8 - Lines 419-421, 424-428: these statements need appropriate referencing.

The reviewer is right, and missing references were added accordingly (page 10-11, lines 446 and 454).

9 - Line 541: because enzymes are proteins, they do not need separate mentioning.

The reviewer is right and the term ‘enzymes’ was deleted to avoid redundancy.

Reviewer 3 Report

The article by Cassani et al. is well written, the comprehensive properties of selected types of algae are described here. Methodologically, everything is well described here and the graphics of the article are satisfactory. In conclusion, the authors summarize that seaweed can be used as a functional organic food or biofuel. However, this fact has long been known.

Overall, I rate the article positively, however, given the content, I would recommend it for further review by Foods MDPI journal.

Author Response

Dear reviewer,

Thank you for your important comments on the improvement of the manuscript. We want to apologize for the delay in responding to the comments. Hope this new version has the expected quality.

Please, find next the responses to your comments. We deeply revised the manuscript and responded to each issue raised by each of the reviewers in ‘red color’.

Changes in the revised version of the manuscript including text, figures and tables were done with the Track & Changes tool and newly added references were highlighted in yellow.

Yours sincerely,

The authors

The article by Cassani et al. is well written, the comprehensive properties of selected types of algae are described here. Methodologically, everything is well described here and the graphics of the article are satisfactory. In conclusion, the authors summarize that seaweed can be used as a functional organic food or biofuel. However, this fact has long been known.

Overall, I rate the article positively, however, given the content, I would recommend it for further review by Foods MDPI journal.

We deeply appreciate the reviewer’s observation, but we consider that our study matches the IJMS's scope and offers a relevant and appropriate topic for this journal. The present study deals with the macroalgae valorization in terms of nutritional characterization, thermochemical properties and bioactive profile with interest in biology, and chemistry fields. From the chemical perspective, the nutritional composition of eight seaweed species (lipids, fatty acids, proteins, carbohydrates, minerals) is well described and discussed. In addition, the thermochemical conversion of such species to potentially produce biofuel was comprehensively analyzed (such application is not largely studied in macroalgae). From the biological point of view, bioactive properties such as antioxidant activity (evaluated through several assays), cytotoxic and anti-inflammatory activities of the selected seaweeds were investigated and results for that section can be promising for the food, cosmeceutical, and nutraceutical area. Thus, our study contributes to expanding knowledge about opportunities arising from the valorization of seaweeds, most of them still largely unexplored.

Reviewer 4 Report

The manuscript "Thermochemical characterization of eight seaweed species and evaluation of their potential use as an alternative for biofuel production and source of bioactive compounds" addresses a relevant and appropriate topic for this journal, however, the authors should make the corrections suggested below.

Corrections needed:

line 421/43 - The most widely consumed species belong to the Undaria (wakame)(Ochrophyta, Phaeophyceae), Porphyra (nori)(Rhodophyta) and Laminaria (kombu)(Ochrophyta, Phaeophyceae) genera [4].

line 83/84 - and Saccharina latissima (Linnaeus) C.E. Lane, C. Mayes, Druehl & G.W. Saunders [formerly Laminaria saccharina (L.) Lamouroux]

Figure 1.

Important Note: As a phycologist myself, and also a connoisseur of the Galicia flora, I can say without any doubt that the identification of "Lechuga de Mar" and "Nori" are not correct.
In the first case it is Ulva rigida (UR) and in the second case the photograph does not correspond to a Porphyra purpurea (PP).
Thus, the authors must correct in this figure the name of the corresponding species "Lechuga de Mar", and in the second case they must replace the image with one effectively of Porphyra purpurea.
If in fact the seaweed that Algamar has supplied you with is Ulva lactuca, the authors must exchange the image for one of the Ulva lactuca species.
In the case of Kombu Real, the valid name of the species is Saccharina latissima (SL).

Table 1
Replace LS, Laminaria saccharina for SL, Saccharina latissima

line 457/458 - ... (except Laminaria saccharina) were ...

Table 2
Replace LS, Laminaria saccharina for SL, Saccharina latissima

line 489 - an excessive consumption of L. ochroleuca and S. latissima could lead to undesirable

Table 3
Replace LS, Laminaria saccharina for SL, Saccharina latissima

line 500 - were seen and ranged from 5.9% for C. tomentosum to 14.9% for S. latissima.

Table 4
Replace LS, Laminaria saccharina for SL, Saccharina latissima

Figure 2
...Saccharina latissima; g...

line 595 - ... S. latissima,

Table 5
Replace LS, Laminaria saccharina for SL, Saccharina latissima

Table 6
Replace LS, Laminaria saccharina for SL, Saccharina latissima

Author Response

Dear reviewer,

Thank you for your important comments on the improvement of the manuscript. We want to apologize for the delay in responding to the comments. Hope this new version has the expected quality.

Please, find next the responses to your comments. We deeply revised the manuscript and responded to each issue raised by each of the reviewers in ‘red color’.

Changes in the revised version of the manuscript including text, figures and tables were done with the Track & Changes tool and newly added references were highlighted in yellow.

Yours sincerely,

The authors

The manuscript "Thermochemical characterization of eight seaweed species and evaluation of their potential use as an alternative for biofuel production and source of bioactive compounds" addresses a relevant and appropriate topic for this journal, however, the authors should make the corrections suggested below.

We are very grateful for the reviewer’s observation.

Corrections needed:

1 - line 421/43 - The most widely consumed species belong to the Undaria (wakame) (Ochrophyta, Phaeophyceae), Porphyra (nori)(Rhodophyta) and Laminaria (kombu)(Ochrophyta, Phaeophyceae) genera [4].

The reviewer is right. That sentence was corrected (page 1-2, lines 42-44).

2 - line 83/84 – and Saccharina latissima (Linnaeus) C.E. Lane, C. Mayes, Druehl & G.W. Saunders [formerly Laminaria saccharina (L.) Lamouroux]

We appreciate the reviewer’s observation and Laminaria saccharina was replaced by Saccharina latissima throughout the revised version of manuscript including text, figures, and tables.

3 - Figure 1.

Important Note: As a phycologist myself, and also a connoisseur of the Galicia flora, I can say without any doubt that the identification of "Lechuga de Mar" and "Nori" are not correct. In the first case it is Ulva rigida (UR) and in the second case the photograph does not correspond to a Porphyra purpurea(PP). Thus, the authors must correct in this figure the name of the corresponding species "Lechuga de Mar", and in the second case they must replace the image with one effectively of Porphyra purpurea. If in fact the seaweed that Algamar has supplied you with is Ulva lactuca, the authors must exchange the image for one of the Ulva lactuca species. In the case of Kombu Real, the valid name of the species is Saccharina latissimi (SL).

The reviewer is right, and we are grateful for such an observation. Certainly, this comment has been very valuable to improving our manuscript. After carefully checking with the seaweed supplier, we have realized that the species used in this study was Ulva rigida. Thus, Ulva lactuca was replaced by Ulva rigida throughout the entire revised version of the manuscript. In addition, the picture for Porphyra purpurea has been changed by the correct one. The valid name of Kombu Real, Saccharina latissima (SL), was introduced in the revised manuscript.

4 - Table 1

Replace LS, Laminaria saccharina for SL, Saccharina latissima

We are very grateful for the reviewer’s observation and such replacement was done.

5 - line 457/458 - ... (except Laminaria saccharina) were ...

We are very grateful for the reviewer’s observation and such replacement was done.

6 - Table 2

Replace LS, Laminaria saccharina for SL, Saccharina latissima

We are very grateful for the reviewer’s observation and such replacement was done.

7 - line 489 - an excessive consumption of L. ochroleuca and S. latissima could lead to undesirable

We are very grateful for the reviewer’s observation and such replacement was done.

8 - Table 3

Replace LS, Laminaria saccharina for SL, Saccharina latissima

We are very grateful for the reviewer’s observation and such replacement was done.

9 - line 500 - were seen and ranged from 5.9% for C. tomentosum to 14.9% for S. latissima.

We are very grateful for the reviewer’s observation and such replacement was done.

10 - Table 4

Replace LS, Laminaria saccharina for SL, Saccharina latissima

We are very grateful for the reviewer’s observation and such replacement was done.

11 - Figure 2

...Saccharina latissima; g..

We are very grateful for the reviewer’s observation and such replacement was done.

12 - line 595 - ... S. latissima,

We are very grateful for the reviewer’s observation and such replacement was done.

Table 5

Replace LS, Laminaria saccharina for SL, Saccharina latissima

We are very grateful for the reviewer’s observation and such replacement was done.

Table 6

Replace LS, Laminaria saccharina for SL, Saccharina latissima

We are very grateful for the reviewer’s observation and such replacement was done.

Round 2

Reviewer 3 Report

Authors improved manuscript, thus I recommend to publish it in current version.